# Violaceous Lesions on the Leg: What Else Apart from Kaposi Sarcoma? Differential Diagnosis with a Narrative Review of the Literature

**Alessandro Pileri** [1,2,†], **Gionathan Orioni** [1,2,†], **Corrado Zengarini** [1,2,*], **Vieri Grandi** [3], **Bianca Maria Piraccini** [1,2] and **Valeria Gaspari** [2]

1   Dipartimento di Scienze Mediche e Chirurgiche, Università di Bologna, 40138 Bologna, Italy
2   Dermatology Unit, IRCCS Azienda Ospedaliero-Universitaria di Bologna, 40138 Bologna, Italy
3   Section of Dermatology, Department of Health Sciences, University of Florence, 50100 Florence, Italy
*   Correspondence: corrado.zengarini@studio.unibo.it; Tel.: +39-0512144838
†   These authors equally contributed to this work.

**Abstract:** With this work, we aimed to review the principal benign and malignant tumors (including vascular, keratinocytic/epidermal, melanocytic, hematopoietic, and lymphoid origin), primarily affecting the leg's skin. The lesions' location can also help focus on a spectrum of differential diagnoses in clinical practice. All the diseases present the same clinical presentation characterized by erythematous to violaceous nodules. Despite the same clinical presentation, each disease's prognostic outcome and therapeutic management can be somewhat different. Since clinical diagnosis may sometimes be challenging, histology and immunohistochemistry play a fundamental role in recognizing and staging these types of lesions. Molecular studies can help to determine the exact nature of lesions with no specific characteristics. Kaposi's sarcoma is an angioproliferative neoplasm that typically occurs in the lower limbs and can enter into differential diagnosis with several other rarer skin diseases. The principal differential diagnosis concerns primary cutaneous lymphomas, of which mycosis fungoides represent the most frequent primary cutaneous T-cell lymphoma. Other rare forms include primary cutaneous B-cell lymphomas, which can be divided into indolent and aggressive forms, such as the primary cutaneous diffuse large B-cell lymphoma, leg type, and lymphomatoid papulomatosis (LyP). In the case of indolent lesions, skin-directed therapies, limited-field radiotherapy, and surgical approaches can be good options. At the same time, different management, with systemic chemotherapy and allogenic bone marrow transplant, is required with aggressive neoplasms, such as blastic plasmacytoid dendritic cell neoplasia or advanced mycosis fungoides. The dermatologist's role can be crucial in recognizing such diseases and avoiding misdiagnosis, giving the pathologist the correct clinical information for an accurate diagnosis, and starting the suitable therapy.

**Keywords:** primary cutaneous lymphomas; Kaposi sarcoma; leg; nodular tumors

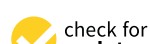



## 1. Introduction

The morphology and color of skin lesions are sometimes crucial in guiding dermatologists toward the correct diagnosis. The lesions' location can also help focus on a spectrum of differential diagnoses in clinical practice. However, the myriad of differential diagnoses in the case of the presence of multiple violaceous lesions on the legs can confuse even an experienced dermatologist. The present paper analyzes the spectrum of possible diseases that can present as violaceous nodules or plaques on the leg, with a skin genesis focusing on the most common and uncommon diagnosis, a quick overview of the main differential diagnoses is shown in Table 1.

**Table 1.** Rapid chart of clinical characteristics, diagnostic criteria, differential diagnosis, prognosis and treatment of the main neoplastic, violaceous lesions of the legs.

| | Presentation | Diagnostic Approach | Princiapl Differential Diagnosis | Aggressivity | Possible Therapies |
|---|---|---|---|---|---|
| Basal cell carcinoma | Single | Dermoscopic and Histological diagnosis | Squamous cell carcinoma and amelanotic melanoma | Locally invasive | Topical Therapy, surgical excision |
| Squamous cell carcinoma | Single | Dermoscopic and Histological diagnosis | Basal cell carcinoma and amelanotic melanoma | Aggressive | Topical Therapy, surgical excision, radiotherapy |
| Kaposi sarcoma | Multiple | Histological and immunohistochemistry | Lymphomatoid papulosis | Indolent or aggressive | Surgical excision, radiotherapy, systemic drugs |
| Pyogenic granuloma | Single | Histological | Amelanotic melanoma | Indolent | Topical beta-blockers, topical steroids, cryosurgery or electrodesiccation |
| Amelanotic melanoma | Single | Histological | Pyogenic granuloma | Aggressive | Surgical excision |
| Lymphomatoid papulosis | Multiple | Histological | Kaposi sarcoma and CTCL | Indolent | Include topical creams, laser therapy or surgical excision |
| Anaplastic large cell lymphoma | Single | Histological | Other CTCL | Indolent or aggressive | Chemotherapy, radiotherapy, rarely surgical excision |
| Primary cutaneous diffuse large b-cell lymphoma, leg type | Multiple | Histological and immunohistochemistry | Other CBCL | Aggressive | Chemoimmunotherapy |
| Primary cutaneous marginal zone lymphoma | Single | Histological and immunohistochemistry | Other CBCL | Indolent | Field radiotherapy, surgery, chemotherapy |
| Primary cutaneous follicle center lymphoma | Multiple | Histological and immunohistochemistry | Other CBCL | Indolent | Field radiotherapy, surgery, chemotherapy |
| Mycosis fungoides | Single to multiple | Histological, immunophenotype and molecular characterization | Other CTCL, eczematous dermatitis, psoriasis | From indolent to aggressive | Phototherapy, systemic regimens, chemotherapy, radiotherapy |
| Blastic plasmacytoid dendritic cell neoplasm | Single to multiple | Histological and immunohistochemistry | Amelanotic melanoma and other CTCL/CBCL | Aggressive | Chemotherapy, bone marrow transplant |

## 2. Kaposi Sarcoma

Kaposi sarcoma (KS) is an angioproliferative neoplasm related to human herpes virus 8 (HHV8). Five clinical variants have been described, ranging from Classic KS (CKS), African endemic KS, immunosuppressed-related form, KS in HIV-negative men who have sex with men and AIDS-related KS [1–5]. The incidence of classic KS is higher in Mediterranean countries, and the overall Italian incidence rate has been estimated to be 1/100,000 in men and 0.4/100,000 in women. Iatrogenic KS is commonly detected in organ transplant recipients (OTR), with an increased risk estimated to be from 100 to 500 times higher than in the general population. Regarding the clinical aspects, skin manifestations can range from pink to violaceous papules, patches, and ulcerated nodules (Figure 1a). CKS usually involves the legs, and lymphedema is the most common cutaneous complication. Oral, genital or conjunctival involvement can be detected at physical examination [6], and in advanced stages, lymph node and visceral (bladder, lung and kidney) involvement are possible. CKS has an indolent clinical behavior, while the remaining forms are more

aggressive and easily involve lymph nodes or visceral organs [2]. Besides CKS, which usually affects the sixth decade, patients of any age can be affected, and pediatric cases have been recorded [7,8]. Histology is crucial for KS diagnosis. Skin lesions show similar features and patches with a perivascular infiltrate of scattered plasma cells and lymphocytes. A proliferation of oddly-shaped endothelial cell-lined vascular channels behind the existing blood vessels is a clue for KS diagnosis. Such channels become more prominent in patch lesions along with the presence of CD31+ and CD34+ spindle cells. In nodular lesions, a dermal mass of spindle cells is visible with light microscopy along with extravasated red blood cells.

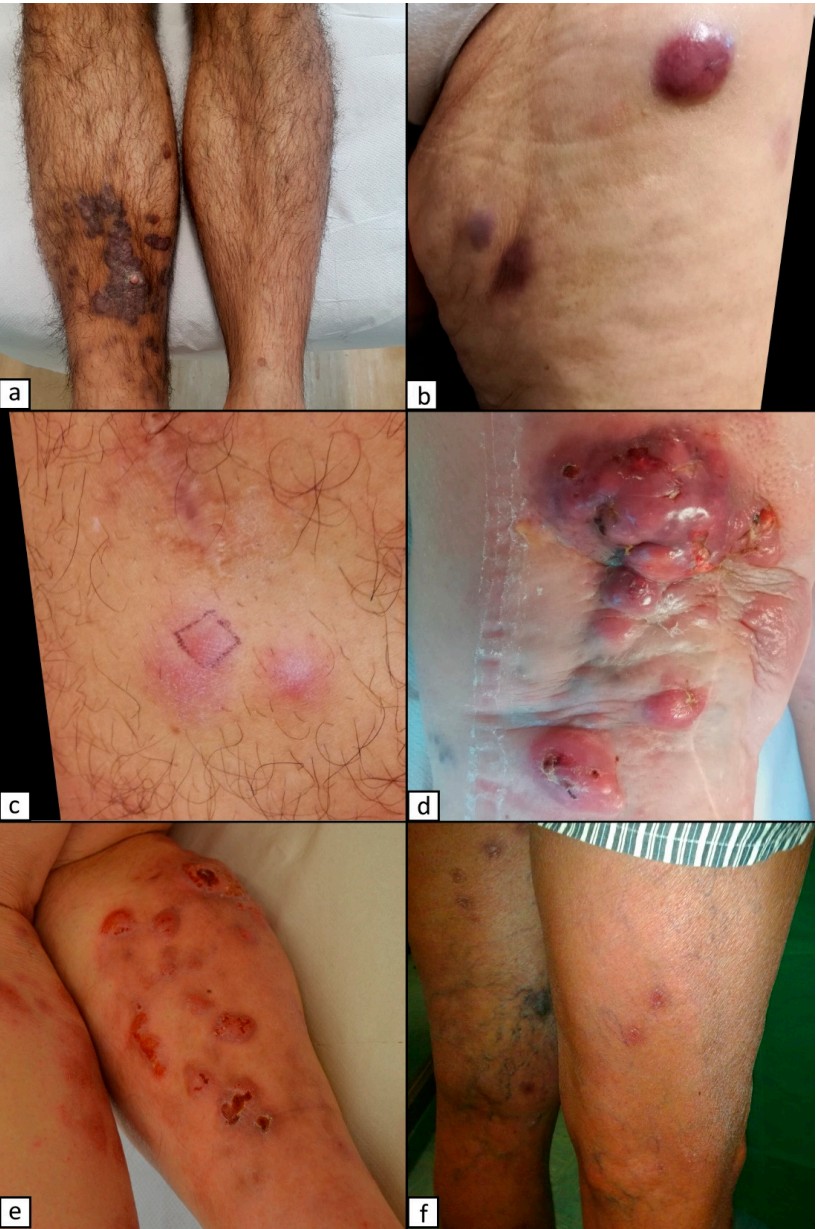

**Figure 1.** Erythematous to purplish nodules and plaques located on the leg in different diseases, (**a**) a purplish plaque and nodules along with monoliteral edema of the leg in a Kaposi sarcoma case, (**b**) purplish nodules in a primary cutaneous marginal zone lymphoma, (**c**) erythematous subcutaneous nodules in a primary cutaneous follicle center lymphoma; (**d**) multiple nodules along with crusty lesions in a primary cutaneous diffuse large cell lymphoma; (**e**) multiple ulcerated nodules and erythematous plaques in a mycosis fungoides; (**f**) erythematous nodules on the legs in a blastic plasmacytoid dendritic cell neoplasia case.

In immunohistochemistry, staining against HHV-8 results in positive staining in spindle cells. Scheduled treatments depend on the clinical variant, KS extension and immune system status. Dermoscopy of KS can be helpful, and polychromatic color change (Rainbow pattern), collarette sign, white lines, white clouds, and serpentine vessels have been related to KS, especially in the case of nodular lesions [9]. The Italian recommendations [6] suggest treating with local therapy, (mainly intralesional vincristine or silver nitrate cauterization) as lesions can negatively affect patients' quality of life in case of indolent KS lesions. Aggressive, diffuse and visceral KS should be managed with systemic treatments such as vinblastine alone, vinblastine associated with bleomycin, or paclitaxel [6] (Brambilla IJDV 2021). In immunosuppressed patients, medication changes usually lead to KS regression [2,10,11]. In HIV-related forms, pegylated liposome doxorubicin or paclitaxel are commonly used along with ART (Antiretroviral Therapy) [6,12,13].

## 3. Pyogenic Granuloma

Lobular capillary hemangioma, or pyogenic granuloma (PG), can be defined as a benign vascular proliferation of the skin and mucous membranes. PG pathogenesis is still unknown, while the disease is usually characterized by a solitary pink to red dome-shaped nodule. In histology, PG presents as a vascular proliferation organized in lobules, along with fibrous septations, inflammation, and edema resembling granulation tissue with no or rare extravasated red blood cells. PG usually involves the fingers, but it can be virtually present on any body part, including mucous membranes [14,15]. Some cases can be challenging, and PG-like KS has been described in the literature [16]. In those cases, dermoscopy of the lesions can be helpful, mainly consisting of reddish structureless (homogeneous) areas along with white intersecting lines, ulceration and crusts and peripheral collarette scale [17–19]. Topical beta-blockers, topical steroids, cryosurgery or electrodesiccation can be the most common therapeutic choice [14].

## 4. Amelanotic Melanoma

Amelanotic melanoma (AM) is a rare form of melanoma accounting for 2–8% of all malignant melanoma (MM) diagnoses [20]. AM's mean age is over 50 years, and the male/female ratio varies from 0.5 to 4, according to literature data [21]. Patients with type I skin or sun-sensitive phenotype are more likely to develop AM. The lack of pigmentation and the clinical presentation as a papule or a reddish nodule resembling benign melanocytic or non-melanocytic lesions, the absence of the most common features of the pigmented melanoma (PM) make the definition of "the great masquerader" appropriate, to highlight that AM diagnosis is challenging [22]. From a histological point of view, AMs are more commonly nodular melanoma, acral lentiginous melanoma, and desmoplastic melanoma rather than PM. They also present higher Breslow thickness, mitotic rate, more frequent ulceration, and higher tumor stage than PM. Therefore, AM prognosis is worse than PM due to the abovementioned features and the higher proportion of delayed diagnosis [20,21]. Kelly et al. [23] have proposed the EFG criteria (elevated, firm, growing for more than one month) to avoid under-detection of AM cases. AM can present in any site of the body, including the legs. Dermoscopy in AM can be a helpful tool, and several dermoscopic features have been related to AM, including vascular structures [20,22,24]: polymorphic vessels, blue-white veil, multiple blue-grey dots, milky red-pink areas, asymmetric shape, multiple colors, ulceration, and a scar-like depigmentation. The presence of dotted vessels in combination with vessels with different morphology (i.e., linear irregular, looped, or serpentine vessels) may alert clinicians. Surgical removal of the lesions is mandatory.

## 5. Pigmented Basal Cell Carcinoma (SCC)

Pigmented basal cell carcinoma (BBCC) is a subtype of basal cell carcinoma (BCC) characterized by the presence of pigmentation or dark coloring in the tumor [25,26]. The incidence of pigmented basal cell carcinoma (BBCC) is not well-established, as it is a subtype of basal cell carcinoma (BCC) and most studies do not specifically track the incidence of

pigmented BCCs [25]. However, it is estimated to be less common than non-pigmented BCCs. It is estimated that BCC accounts for around 80–85% of all skin cancer cases. It can resemble a mole or a dark freckle and may be brown, black blue or also violaceus in color. Pigmented BCCs can be more difficult to diagnose than non-pigmented BCCs because they can resemble other skin lesions [27]. They are often raised and may have an irregular border. They are commonly found on sun-exposed areas such as the face, ears, and scalp but can occur on any part of the body, including the legs. The exact reason why a pigmented BCC may develop on the legs is not well understood, but it is thought to be related to exposure to UV radiation from the sun or tanning beds. Exposure to UV radiation is a major risk factor for all types of skin cancer, including BCC. UV radiation damages the DNA in skin cells, leading to mutations that can lead to the development of cancer. The legs are often exposed to UV radiation, especially in individuals who wear shorts or skirts or who have a profession that requires them to work outdoors. Individuals with fair skin, a history of sunburns, or a history of prolonged sun exposure are at an increased risk of developing BBCC on the legs. Additionally, people with certain genetic conditions such as Xeroderma Pigmentosum, Basal Cell Nevus Syndrome, Gorlin–Goltz syndrome are more susceptible to developing BCC.

## 6. Squamous Cell Carcinoma (SCC)

Squamous cell carcinoma (SCC) is one of the most common nonmelanoma skin cancer accounting for 20% of skin cancer. It has been estimated that every year in the United States, 1 million cases are diagnosed with a number of 9000 estimated death [28–30]. SCC incidence is estimated to increase due to the aging population. SCC clinical presentation can vary, ranging from erythematous, scaly, well-demarcated plaque (Bowen's disease) to an isolated red scaly plaque or nodule, typically in a sun-exposed area. SCC can be preceded by actinic keratosis, a pre-malignant condition described as scaly, flesh-colored, pink or brown papules or plaques, often with an erythematous [31]. Ultraviolet (UV) rays have been recognized as the most critical carcinogen in SCC pathogenesis [32,33]. In histology, SCC may vary from well to poorly differentiated. The former exhibits interconnecting follicular infundibular type squamous epithelium and rare or absent mitosis. Poorly differentiated SCC shows neoplastic cells featuring great cytological atypia and it is difficult to determine a keratinocyte lineage [34]. In dermoscopy, the combination of clustered dotted/glomerular vascular patterns and hyperkeratosis, seen as discrete yellow scales, has previously been shown to achieve a 98% diagnostic probability for SCC. A subset of SCC (10% of all SCC) can affect the lower extremities (LE) and shows some peculiarities. LE SCC commonly affects females and, surprisingly, is more common than basal cell carcinoma (BCC) [32]. This feature suggests that local immunosuppression may play a role in LE SCC pathogenesis. Indeed, the higher rate of SCC on BCC is a common feature in immunosuppressed patients (i.e., organ transplant recipients). Theoretically, the leg can be considered an area prone to local immunosuppression that can help the onset of neoplastic lesions, as observed in KS. Moreover, minor trauma or UV rays can be related to LE SCC pathogenesis. Most LE SCC can present as nodular erythematous to crusty lesions with no features of aggressive SCC.

## 7. Lymphomatoid Papulosis (LyP)

Lymphomatoid papulosis is a rare, benign disorder of the skin that is characterized by the presence of multiple, small, flesh-colored or reddish papules (small, raised bumps) on the skin [35]. The papules are usually less than 5 mm in diameter and they are typically found on the legs [36]. Lymphoid papulosis is thought to be a type of reactive lymphoid [35] cells (white blood cells that are important in the immune system) in the skin. Based on prognosis and histological features there are three main subtypes (A, B, C) of lymphomatoid papulosis (LyP): recently, histopathologic classification has divided this disorder into six subtypes, adding other three minor variants of disease (D, E, F) [37]. Type A is the most common subtype and is characterized by the presence of small, red to purple, scaly papules or nodules on the skin. This subtype is considered to have a minimal risk of malignant

transformation and has an excellent prognosis. Type B is characterized by the presence of larger, more elevated, and more numerous papules or nodules, that can be itchy or painful. This subtype is considered to have a moderate risk of malignant transformation, and the prognosis is generally good. Type C is characterized by the presence of large, raised, and often ulcerated plaques, often accompanied by systemic symptoms such as fever, night sweats and weight loss. This subtype is considered to have the highest risk of transformation in cutaneous T-cell lymphoma (CTCL) and the prognosis is generally poor. The exact cause of lymphoid papulomatosis is not known, but it is thought to be related to an underlying infection or immune disorder. It is important to note that Lymphomatoid papulosis (LyP) is considered a rare, benign disorder that may evolve into a CTCL, but most cases of LyP do not progress to CTCL. Regular follow-up by a dermatologist is important for any LyP patient, and a biopsy may be needed to confirm the diagnosis and exclude other skin conditions with a similar clinical presentation. However, the papules can be cosmetically unappealing, and some people may choose to have them removed for cosmetic reasons. Treatment options include topical creams, laser therapy or surgical excision.

## 8. Anaplastic Large Cell Lymphoma (ALCL)

Lymphoid Anaplastic large cell lymphoma (ALCL) of the skin is a type of non-Hodgkin lymphoma, which is a cancer that originates in the cells of the lymphatic system. ALCL is a rare subtype of T-cell lymphoma that can arise in various body locations, including the skin. It typically presents as a single, painless, rapidly growing red or purple nodule, often with a visible blood vessel, on the skin of the head, neck, or extremities [38]. It can also occur in other organs such as the lungs, breast and testicles. ALCL is usually treated with a combination of chemotherapy and radiation therapy, and in some cases, surgery may be required. The prognosis for ALCL of the skin is generally good, with a high cure rate that can reach up to 80% at 5 years of follow-up [39]. However, if it spreads to other organs, it can be more aggressive and harder to treat.

## 9. Primary Cutaneous Marginal Zone Lymphoma

Primary cutaneous marginal zone lymphoma (PC-MZL) is an indolent lymphoma accounting for 30% of all cutaneous B-cell lymphoma (CBCL). PC-MZL has been defined as an indolent proliferation of small B-cells, plasma cells and lymph-plasmacytoid cells in the last WHO Classification [40–42]. The neoplasm usually presents as an isolated lesion or multiple erythematous to violaceous nodules, papules or plaques on the extremities (Figure 1b). Histologically, the disease is characterized by nodular infiltration of small lymphocytic cells, lymph-plasmacytoid cells, mature plasma cells, and reactive germinal centers with macrophages, sparing the epidermis and involving the dermis and subcutaneous fat [40–43]. In immunohistochemistry, the neoplastic proliferation expresses CD20, CD79a, and BCL-2 and are monotypic for kappa or lambda light chains [44]. Staining for CD5, CD10, CD23, and CyclinD1 is absent, as well as BCL-6 and PD-1. Cases involving the leg usually present the same indolent clinical course (5-year OS > 95%) of all PC-MZL, which rarely spreads to internal organs. PC-MZL treatment can range from a wait-and-see strategy for asymptomatic lesions to limited-field radiotherapy or the surgical removal of lesions [40,45–47]. No prospective clinical trials have been proposed, so no standardized protocols are available.

## 10. Primary Cutaneous Follicle Centre Lymphoma

Primary cutaneous follicle center lymphoma (PC-FCL) is a malignant proliferation of neoplastic follicle center cells (centrocytes and centroblasts) featuring an indolent clinical behavior [48,49].

PC-FCL is the most common CBCL, accounting for 50–60% of all cases. It commonly affects the male gender with a peak of incidence in the fifth decade of life [41,50–54]. At clinical examination, the disease can present as erythematous to violaceus patches, plaques, or nodules on the head and neck or the back [46,55]. The lesions are usually asymptomatic

and only in rare circumstances do patients complain of an itch or pain [44]. In histology, an infiltrate consists of multilobulated and medium-to-large-sized centrocytes along with a few centroblasts with a varying presence of histiocytes, eosinophils and reactive T-cells. The infiltrate spares the epidermis and presents a perivascular and periadnexal distribution with a growth pattern ranging from follicular to diffuse [55–57]. In immunohistochemistry, B-cell marker lineages such as CD20, CD79a and Bcl-6 are expressed, while the absence of the Bcl-2 molecule is crucial for differentiating PC-FLC from a skin spread of nodal disease. However, in the literature, PC-FLC cases with Bcl-2 expression have been reported with no impact on indolent clinical behavior [58–69]. Although the leg is rarely affected, it can be a sign of a worse clinical outcome (Figure 1c), while PC-FCL usually has an indolent course with a five and 10-year overall survival (OS) of 95 and 88%, respectively [70]. Limited field radiotherapy is the first-line treatment of choice, while alternative options can be the surgical removal of the lesion, intralesional rituximab or high potency steroids [71–73].

## 11. Primary Cutaneous Diffuse Large B-Cell Lymphoma, Leg Type

Primary cutaneous diffuse large B-cell lymphoma, leg type (PC-DLBCL-LT), is a rare and aggressive type of CBCL, usually affecting females in their seventh decade of life [41,44,74]. The disease is rare and accounts for 10–20% of all CBCL [50]. On clinical examination, PC-DLBCL-LT presents as fast-growing violaceous to purplish nodules involving one or both legs (Figure 1d). Cases featuring an isolated lesion or with non-leg involvement have been described [75,76]. At histology, monomorphous infiltrate consisting of neoplastic elements centroblast or immunoblast-like cytology sparing the epidermis with a sheet-like growth pattern can be observed. The neoplastic elements show a high proliferation rate and mitotic figures. At immunohistochemistry the infiltrate expresses CD20, Bcl-2, IRF4/MUM1, FOXP1, MYC, cytoplasmic IgM, CD79a and monotypic immunoglobulin. Other markers, such as Bcl-6 are commonly expressed, while CD10 is negative. Although PC-DLBCL-LT is an aggressive disease, the introduction of chemoimmunotherapy (rituximab plus multiagent chemotherapy, CHOP (cyclophosphamide, doxorubicin, vincristine, and prednisone) or CHOP-like schemes) has slightly changed the global overall survival, which is currently estimated at 5 years in 70% of the patients [74]. However, negative prognostic factors, such as localization on the leg, multiple lesions and age over 75 years, can negatively portend the prognosis.

## 12. Mycosis Fungoides

Mycosis fungoides (MF) is the most common cutaneous lymphoma (CTCL), usually affecting the adult male [44]. The disease is characterized by patch lesions that can precede the appearance of plaques and tumor lesions [77]. Every single clinical presentation is related to peculiar histology. Early phases can be challenging to diagnose [46]; in plaque lesions, the infiltrate is more conspicuous. Tumor lesions show the presence of blasts infiltrating the fat tissue; early stages show scattered neoplastic cells infiltrating the epidermis and intermingled with reactive inflammatory cells. In the early stages, the partial loss of T-cell markers, such as CD5 or CD7, and the predominance of CD4 on CD8 (or vice versa, especially in pediatric cases) can lead to MF diagnosis. In advanced phases, the expression of the CD30 molecule is related to a worse clinical outcome [36,78,79]. Early stages are usually treated with skin-directed therapies (phototherapy, high-potency steroid and local chemotherapy); in advanced phases, systemic treatments (i.e., immune response modifiers, whole body radiotherapy and mono/multiagent chemotherapy) are usually scheduled [80,81]. Albeit rare, cases primarily involving the leg have been described [82–84] (Figure 1e). Case reports range from early-stage disease featuring papular or purpuric lesions to advanced disease mimicking chronic venous ulcer. Due to the absence of case series, no conclusions can be drawn as to the possible different clinical outcomes of the disease.

### 13. Blastic Plasmacytoid Dendritic Cell Neoplasm (BPDCN)

BPDCN is a rare and aggressive neoplasm accounting for less than 1% of acute leukemia. The disease usually affects male subjects in the sixth decade of life, and the median survival is 14 months [85–87]. However, patients younger than 50 years and even pediatric cases have been reported [88–91]. Usually, the skin is the first site of disease involvement, although BPDCN can affect the internal organs and circulating blood [92–98]. Patients featuring skin involvement as the first manifestation usually present with a disease that continues to be confined to the skin until rapid leukemic dissemination or visceral spread occurs. In such cases, it has been hypothesized that the skin may be considered a sanctuary organ initially limiting the disease spread [85]. Clinically, skin lesions can range from tiny violaceous papules to massive, ulcerated tumors. Although BPDCN has no preferred site of involvement, most pediatric cases described present as violaceous papules and nodules (at times located on the subcutis) on the leg [90] (Figure 1f). At histology, BPDCN shows a diffuse monomorphous infiltrate consisting of medium-sized blast cells with irregular nuclei, fine chromatin and small eosinophilic nucleoli. The cytoplasm usually appears scant and grey-blue, while mitotic figures can commonly be observed. The neoplastic elements resembling either lymphoblasts or myeloblasts predominantly infiltrate the dermis, sparing the epidermis and adnexal structures, eventually extending to subcutaneous fat. No angioinvasion or coagulative necrosis is usually detected. At immunohistochemistry, BPDCN diagnosis relies on the expression of the plasmacytoid dendritic cell-associated antigens (CD123 and TCL1), as well as on CD4, CD56 and CD303 positivity. Furthermore, B and T-cells markers are absent as well as myelomonocytic and NK ones. However, in a few instances, staining for CD4 or CD56 can turn out to be negative [99], while in the literature a variable expression in the CD303 marker has been reported [100–103]. Recent studies [86,87] have highlighted that the BPDCN cell of origin may be a resting pDC (plasmacytoid Dendritic Cell) precursor, and the complex karyotype usually observed is related to the aberrant activation of the NF-KB pathway. BPDCN treatment can range from acute myeloid leukemia (AML) or acute lymphoblastic leukemia (ALL) regimens followed by an allogenic bone marrow transplant [104,105]. Recently, tagraxofusp (SL-401), a CD123-directed cytotoxin consisting of human interleukin-3 fused to truncated diphtheria toxin has become available and clinical trials have observed good clinical results with survival rates of 59% and 52% at 18 and 24 months, respectively [106]. Palliative treatment, such as limited-field radiotherapy, can be a sound choice in patients with co-morbidities [85].

### 14. Conclusions

In the present paper, we aimed to review the principal malignancies primarily affecting the skin and with a prominent leg involvement. All the diseases present the same clinical presentation characterized by erythematous to violaceous nodules. Despite the same clinical presentation, each disease's prognostic outcome and therapeutic management can be somewhat different. Indeed, in KS or in PC-FCL and PC-MZL, limited-field radiotherapy or a surgical approach can be a good option for commonly indolent diseases [41,45,72,80], while the same approach cannot be adopted for the other diseases discussed in the present paper. The opposite "polar end" can be seen in the treatment of BPDCN, where patients require an intensive chemotherapy approach (acute myeloid/lymphoid leukemia regimens) followed by an allogenic bone marrow transplant [86,105]. Another possible option is the new anti-CD123 monoclonal antibody tagraxofusp (SL-401), followed by bone marrow transplant procedures [106]. In other neoplastic diseases, such as MF or DLBCL-LT, a multidisciplinary approach, such as a combination of radiotherapy and chemoimmunotherapy can be scheduled, but it will not be discussed in detail owing to the topic of our paper [44,50]. An appropriate and timely diagnosis can be crucial in starting therapeutic procedures for those malignancies. Therefore, the dermatologist's role can be crucial in recognizing such diseases and avoiding misdiagnosis, giving the pathologist the correct clinical information for an accurate diagnosis, and starting suitable therapy.

**Author Contributions:** Conceptualization: A.P., G.O. and V.G. (Valeria Gaspari); Data curation: G.O. and C.Z.; Formal analysis: A.P., G.O. and C.Z.; Methodology: G.O. and C.Z.; Project administration: G.O., C.Z. and B.M.P.; Software: G.O. and C.Z.; Validation: B.M.P. and V.G. (Vieri Grandi); Visualization: V.G. (Valeria Gaspari) and B.M.P.; Writing—original draft: A.P., G.O. and C.Z.; Writing—review & editing: G.O. and C.Z. All authors have read and agreed to the published version of the manuscript.

**Funding:** The authors report no involvement in the research by any sponsor that could have influenced the outcome of this work. This research received no external funding.

**Institutional Review Board Statement:** Not required.

**Informed Consent Statement:** Not applicable.

**Data Availability Statement:** Data derived from public domain resources.

**Conflicts of Interest:** The authors certify that there is no conflict of interest with any financial organization regarding the material discussed in the manuscript.

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
