# Peer review of "Violaceous Lesions on the Leg: What Else Apart from Kaposi Sarcoma? Differential Diagnosis with a Narrative Review of the Literature"

_dermato, doi:10.3390/dermato3010005_

Round 1

Reviewer 1 Report

Great job! Awesome paper!

The paper covers the differential diagnosis of KS lesions on the legs. 

The topic is very relevant as there is a lack of studies addressing ks differential diagnosis. 

It provides beautiful pictures and detailed descriptions that will be of aid to clinicians. 

The methodology is sound. 

The conclusions are fully in line with the paper. 

The references are appropriate and cover all meaningful studies on KS

Graphical elements are suitable to the article and assist the reader in absorbing all information.

Author Response

Dear reviewer.

We would like to thank you very much for your appreciation.

Reviewer 2 Report

Is a well-organized work. Basalioma pigmented, lymphomatoid papulosis and ALCL could be added to the list. It would be more immediate for the reader to  have a table where to evaluate whether the neoplasm most frequently occurs in single or multiple lesions, whether the course is indolent or aggressive, and the type of main therapy (radiotherapy, surgical, or chemotherapy). 

Author Response

Dear reviewer, 

Thank you for your valuable appreciation.

We added all the requests you noticed us to the text and a rapid table to assess the principal differential diagnoses and characteristics.

Reviewer 3 Report

An interesting narrative review exploring all possible differential diagnoses of Kaposi sarcoma of the lower legs. I found the article very interesting and suitable for publication after minor revisions. I would also consider adding some infectious diseases, such as bartonellosis to the text.

Author Response

Dear reviewe, thank you for your appreciation and your valuable comment.

It would be very interisting, to add the infectious diseases to the other diagnosis.

But We think that, due to the way the review was set up, we should only onclude the main neoplastic pathologies. If we were suppose to addinfectious pathologies, we would also have to consider vascular ones which unfortunately would cause an elongation of the article which was designed with a different approach.